# Monitoring Eye Movements Depending on the Type of Visual Stimulus in Patients with Impaired Consciousness Due to Brain Damage

**DOI:** 10.3390/ijerph19106280

**Published:** 2022-05-22

**Authors:** Katarzyna Kujawa, Alina Żurek, Agata Gorączko, Roman Olejniczak, Grzegorz Zurek

**Affiliations:** 1Department of Biostructure, Wroclaw University of Health and Sport Sciences, 51-612 Wroclaw, Poland; katarzyna.kujawa@awf.wroc.pl (K.K.); agagoraczko@gmail.com (A.G.); 2Neurorehabilitation Clinic in Wroclaw, 54-530 Wroclaw, Poland; roman.olejniczak@gmail.com; 3Department of Historical and Pedagogical Sciences, Institute of Psychology, University of Wroclaw, 50-529 Wroclaw, Poland; alina.zurek@uwr.edu.pl

**Keywords:** eye tracking, saccades, fixation, neurorehabilitation, disorders of consciousness, brain health

## Abstract

The eyeballs are often the only way to communicate messages as a result of brain damage. However, it is not uncommon for them to become dysfunctional, thus requiring the introduction of appropriate therapy. The trajectory of eye movements (saccadic movements and gaze fixation) during observation of a static and dynamic point presented with an eye tracker was analyzed in the present study. Twelve patients with brain injury of different etiology, with different degrees of consciousness disorders and not communicating through verbal and motor skills, qualified for the study. All participants demonstrated greater eye movement activity when presented with a dynamic task in which they observed a moving point. The findings suggest that effective eye movement therapy must incorporate dynamic stimuli.

## 1. Introduction

The human eyes are an important organ for perceiving the outside world. In order to allow us to accurately see the object in front of us, our eyeballs must be fixated on it. Fixation (150–600 ms) is a small eyeball movement (EM) that is undetectable and invisible to humans, and its function is to hold the image in the central fovea of the retina. If the eye does not make these small movements, the image in front of it would become blurred and fuzzy [1,2,3]. This is because a large number of photoreceptors are located in the relatively small area of the central fovea (1.5 mm) in relation to the entire retina. The central fovea is targeted at the highest acuity of vision. Therefore, in order for the human eye to perceive the image with the best sharpness, the eyeballs must constantly “tune the eye”, so that the light falls at the right angle on the central fovea [4,5]. In addition to minor EMs, there are also rapid movements, called saccades, the duration of which vary from 10 to 100 ms. Saccadic movements occur when changing the object of interest on which the gaze is refocused. They can occur in the form of volitional movement or reflexes: vestibulo-ocular reflex (VOR) or optokinetic reflex (OKR). Both reflexes are used to stabilize the image on the retina [3,6,7].

The cerebellum plays a key role in controlling the current and long-term modulation of EM. It is responsible for the precision of these movements, which translates into optimal image quality as seen by humans. In EM, the most involved parts of the cerebellum are the ocular motor vermis, caudal fastigial nuclei, nodulus and ventral uvula. The ocular motor vermis and caudal fastigial nuclei are crucial for the accuracy and adaptation of saccadic movements (a motor learning process that keeps the amplitude of saccades on target) [8], accuracy (one of the parameters assessing the quality of recorded eye tracking by calculating the average difference between the actual stimulus position and the measured gaze position), [9] and pursuit gain (eye speed to target speed ratio) [10]. Other structures, such as the nodulus and ventral uvula, are involved in the processing of otolithic signals and VOR responses, including velocity storage (responsible for the mechanisms of optokinetic after-nystagmus and vestibular nystagmus) [11,12,13].

Despite the precise system controlling the EM, it is often dysfunctional, including in neurological disorders, because the EM is one of the most complex motor functions for which the central nervous system (CNS) is responsible [14]. Diseases of central nervous structures can result in loss of saccades, fixation or the development of nystagmus [14,15].

The introduction of EM disorder therapy is extremely important in groups of patients after CNS damage because they are mostly unable to communicate using verbal and motor skills and the eyes are often the only way for them to communicate their needs. In addition, the number of patients who survive acute severe traumatic brain injury has increased significantly with the advancement of technology used in intensive care units. Even mild traumatic brain injury (mTBI) can manifest as visual dysfunction, including increased photosensitivity, accommodative and vergence movement deficits, as well as impaired overall health [16]. There are many different intervention methods that can alleviate or improve EM disorders. Depending on the severity of the visual impairment after TBI (Traumatic Brain Injury), vestibular, physical, cognitive, occupational rehabilitation or fusional prism spectacles (in case of diplopia), as well as tinted spectacles (in case of photosensitivity) are used [17]. The literature review shows promising results for vision rehabilitation interventions after mTBI including the use of oculomotor therapy (e.g., targeted EM training exercises) [17,18,19,20,21].

EM process measurement analysis is often used to gain insight into human behavior and perception [2]. The question arises, however, whether knowing the way the eyeballs move, there is a possibility of selecting visual stimuli in such a way as to optimize the therapeutic program for controlling eye movements. The resulting brain–computer interface (BCI) [22] could also be used for diagnostic purposes for non-verbally communicating people [23]. Studies on stimulus type selection have been conducted for a long time. Findings from animal studies in laboratory settings show that biological sensors are more sensitive to detecting dynamic than static stimuli [24]. There are studies available in the literature databases showing how patients with disorders of consciousness (DOC) respond to the type of stimulus (static vs. dynamic) [25,26,27,28,29]. The results collected in these studies are based on a traditional interaction between patient and therapist, not using eye trackers, which analyze eye movements. Thus, they do not offer the possibility of precisely recording the pursuit of eye movements following the presented stimulus. Moreover, they depend on the person performing the examination to assess the patient’s visual performance. Eye-tracking devices, on the other hand, make it possible to precisely register how the patient follows the stimulus displayed to him. 

The purpose of this study was to examine with the use of an ET device how the eyeballs move in DOC patients observing a static and dynamic object. This will answer the question of whether the presentation of the displayed object (static or dynamic) attracts more attention. The practical aspect will be to use this information to take into account the type of stimulus in the rehabilitation of the muscles controlling the EM.

## 2. Materials and Methods

### 2.1. Group Characteristics

The study group consisted of people recruited from the Palliative Care Center in Będkowo, Poland. Data were collected in 2019–2020. The inclusion criteria in the project were: (1) patients not undergoing hospital treatment after completion of standard medical care, (2) consent of the patient’s guardian to participate in the study, (3) lack of verbal and motor communication with the environment, (4) diagnosis of brain damage of different etiology, and (5) at least one functioning eyeball. Project exclusion criteria: lack of consent from the patient’s caregiver to participate in the study, both eyeballs nonfunctional (total damage to both optic nerves), and three failed calibrations. 

Twelve people aged between 26 and 67 years old participated in the study; 7 men and 5 women. The state of consciousness was determined by the doctor using the CRS-R (Coma Recovery Scale Revised). 

The results obtained in the auditory subscale of the CRS-R (Table 1) indicate that auditory function is preserved in all study participants. One patient (P10) scored 0 on the visual subscale of the CRS-R test; however, we decided to include him in further studies due to literature reports indicating a significant error rate in the diagnosis of people with impaired state of consciousness [30]. This is related to the dependence of this assessment on the person making the diagnosis, whereas, in the case of the eye-tracking device, the assessment is made in a way that is independent of the human investigator. 

Approval from the Senate Committee on Research Ethics at the University of Health and Sport Sciences (consent no. 29/2017; date 12 December 2017) was obtained prior to the study. All participants’ supervisors gave written and informed consent for participation and publication of this report in accordance with the guidelines established by the Declaration of Helsinki. Patient characteristics are shown in Table 1, while the participant recruitment process can be found in Figure 1.

### 2.2. Research Tool

The data used in this study were collected using an eye-tracker device (known as C-Eye) at a sampling rate of 30 Hz, an accuracy of 0.4 degrees of visual angle and a velocity threshold of 40 cm/s. The C-Eye device consisted of a 19″ monitor mounted on a metal rack that allowed the screen to be placed in front of the participant’s face. An eye tracker is placed in the lower part of the monitor, emitting infrared radiation (IR), which does not affect the patient’s work with the device in any way (it is invisible to the user). Depending on whether the patient has one or both eyeballs working, the ET tracking mode can be monocular or binocular. For each patient, the distance from the screen surface to the tip of the nose was 50 cm. The position of the monitor was adjusted to the patient’s position during the examination so that the patient’s face was in front of the ET. Prior to the study, a one-point calibration was performed once in the case of each participant, in order to determine the location of the patient’s visual fixation point (the 2D image from the IR camera is processed on the screen of the device). The eye position is estimated by the ET based on the position of the center of the pupil. This position changes with each change in the user’s gaze direction and the two glints that are reflections from the corneal surface of the eye (each glint is a so-called first Purkinje image—P1). These two reflections from the cornea are reference points for the center of the pupil. During the one-point calibration, the patient was asked to observe a red, blinking dot with a white border. The dot moved from the top of the screen and stopped in the middle. The analysis of the patient’s gaze fixation proceeded until the system correctly detected the position of the eyeballs. If the patient was unable to fixate the eyes, the system informed them after 10 s that the calibration had failed, and they were no longer able to proceed with the task. The ET automatically recorded data for each patient, and also exported the recorded information in graphical png. Form (vector lines corresponding to eye movements). Because the eye-tracking system compensates for physical movements (small head movements), no head stabilization was applied to any of the patients. 

### 2.3. Fixation and Saccade Test

Each participant was asked to observe the red dot displayed by the device on the screen (Figure 2). In the static task (ST), the red blinking point remained in the same place for 10 s and was exactly in the center of the monitor. In the dynamic task (DT), the red dot moved randomly to different parts of the screen for 10 s, stopping for two seconds each time. The therapist gave a one-time verbal command of “look at the red dot” before starting the device. The device recorded EM trajectory and gaze fixation (GF). 

### 2.4. Statistical Analysis

For the data obtained, an analysis based on position measures was performed; minimum, maximum, quartiles, and medians were calculated. To assess differences between the number of fixations in ST and DT recorded in and out of the monitor field, the Wilcoxon paired-rank test was used. The compatibility of the distributions of the studied variables was compared using the Friedman rank test (ANOVA). The package used was Statistica, ver.13.1 PL licensed to Biostructure Research Laboratory of Wroclaw University of Health and Sport Sciences, Poland (certificate ISO 9001).

## 3. Results

All EM fixations and trajectories were detected by the device’s built-in software, regardless of whether the patient was gazing outside or within the monitor. The duration of the fixation was between 150 and 600 ms; this time was considered the fixation point and was automatically assigned a number by the program. The EM trajectories are shown by the blue lines. Figure 3, Figure 4, Figure 5 and Figure 6 show the images generated from the device, after completing the tests. Each column shows the tasks for one patient: a static task (top image) and a dynamic task (bottom image). The images contain only the fixations recorded in the monitor field.

Analysis of the EM trajectories for ST and DT shows that DT was performed by all patients with higher accuracy, i.e., the EM trajectories ran closer to the tracked point. At the same time, in ST, only three subjects (P4, P10, P12) had EM trajectories intersecting the observed point.

In dynamic tasks, most patients (seven to eight subjects) were more likely to hold their gaze near the object they were observing. A summary of the number of fixations in ST and DT is shown in Table 2.

Using the Wilcoxon paired-rank order test, it was checked whether there were significant differences between the number of fixations in static and dynamic tasks in the monitor area and outside of it. Such differences were found only between the number of fixations out of the monitor registered in static and dynamic tasks (Table 3). In every other situation (static task—fixation in the monitor and off-monitor, dynamic task—fixation in the monitor and off-monitor, static vs. dynamic task—fixation in the monitor) statistically significant differences occurred.

## 4. Discussion

There are no available studies to date showing how the type of stimulus displayed (by use of ET) to patients with disorders of consciousness affects their interest in observing that stimulus. However, there are studies on how patients respond to personalized and non-personalized visual stimuli. For this purpose, the CRS-R scale was used, which is an effective tool used to diagnose the state of consciousness of patients, allowing differentiation of these states (unresponsive wakefulness syndrome, minimal consciousness, and emergence from minimal consciousness) [31,32]. One of the areas examined in the CRS-R is motor performance, with “Functional Use of Objects” being its component; it is crucial in differentiating between minimal consciousness state (MCS) and emergence from minimal consciousness state (eMCS). In the light of research, it appears that what a patient looks at (personalized versus non-personalized stimuli) when examined using the modified CRS-R is important in correctly diagnosing them [33,34]. An experiment conducted in a group of 21 post-coma patients diagnosed with MCS demonstrates the role of showing the appropriate object to the patient. Functional Item Use was assessed by using personalized items (previously used by patients in activities of daily living), and non-personalized items that were shown in random order. They found that five of the 21 subjects scored higher (eMCS), as long as they were shown personalized items [33]. A similar study was conducted in 2018 by Stenberg, Godbolt, and Möller in a group of patients with impaired consciousness, who were subjected to consciousness assessment using the CRS-R. To compare the patients’ responses, the study swapped the stimuli used from neutral to more personal to the patients. Compared to neutral stimuli, pictures of people close to the patients generated significantly more visual fixations. Stenberg et al. thus conclude that visual stimuli with personal meaning can increase the number of visual fixations compared to the neutral stimuli used in the current standard of consciousness assessment (CRS-R) [34].

The present study analyzed how patients with impaired consciousness resulting from brain injury observe the displayed point on the screen and which visual stimulus (static or dynamic) arouses more interest in what is happening on the monitor, as expressed by their EM control. Based on the presented results, it can be clearly concluded that the visual dynamic stimulus attracted the patient’s attention more. 

By using static and dynamic stimuli in our study, it was assumed that the way patients perceive them (and therefore their response in the form of EM) may be related to the different processing of static and dynamic stimuli by the brain. This may be suggested by the results of a 1983 study involving a female patient who suffered bilateral posterior brain damage. The conclusions of this paper show that motion vision (dynamic stimuli) is a distinct visual function that depends on neural mechanisms outside the primary visual cortex [35,36]. Thus, we assume that patients in our study showed greater EM activity in DT due to the brain’s undisturbed processing of moving stimuli. Furthermore, the literature suggests that global motion perception is a higher-order function, and there is clear segregation between global motion sensitivity and static forms [37].

We also hypothesized that the observed greater EM activity in DT may result from a delayed response to a stimulus, which is often present in individuals after brain injury. The dynamic task was always presented to the patients as the second in the sequence, so that the subjects, due to coexisting cognitive disorders, could respond with a delay to the command given by the therapist. It can therefore be assumed that patients who remembered the command from ST, responded more precisely to the stimulus only during DT. The way humans respond to a stimulus is represented by a simple behavioral scheme (stimulus–response). In this scheme, stimuli reaching the human brain are subjected to conscious analysis, followed by a decision to respond to them or not. The process of moving from thinking to acting was described by Fuster (2000) as the peak of the “perception-action” cycle. According to the described model, the frontal lobe neuronal networks that represent motor or executive memories are likely to be the same networks that collaborate with other cerebral structures in the temporal organization of behavior [38,39]. The process of responding to a stimulus is prolonged following the brain damage [40].

UWS patients in whom multiple gaze fixations and saccadic movements were observed were particularly notable in our study. It is commonly believed that this is a group of patients who are unaware and unable to communicate with their environment [41,42,43]. However, about 15% of these patients show signs of hidden awareness when examined by functional magnetic resonance imaging (fMRI) or EEG, which is known as cognitive-motor dissociation (CMD) [44,45]. Cognitive motor dissociation is also known as covert consciousness characterized by volitional brain activity [46]. We are increasingly able to detect signs of consciousness in UWS patients, which gives us the opportunity to develop appropriate therapies aimed at improving communication with this patient group.

We expect that the incorporation of dynamic stimuli into a rehabilitation program for people following brain damage will be beneficial, not only because of the potential to improve EM control but also to improve the interaction between the patient and the eye tracker. Visual problems also impede clinical assessment of the patient, which can lead to, e.g., misdiagnosis of the consciousness level and depriving the patient of the ability to communicate with the environment. Objective measurements of awareness using an ET can reduce the risk of misdiagnosis and enable patients to communicate their needs [47,48]. The results of our study also highlighted the importance of applying an appropriate stimulus to the DOC patient’s ability to improve contact, thus suggesting the inclusion of eye-tracking in EM rehabilitation. Further research can be carried out to develop ET-based interventions to facilitate ocular function and improve the quality of life in patients who suffered a brain injury. In addition, research with a larger group of participants is needed to confirm the findings of our project.

## 5. Conclusions

The results of the present study indicate that dynamic visual stimuli, compared to static stimuli, have the potential to generate more responses in DOC patients. Based on the data, a moving stimulus is more interesting to the patient. Regardless of which theory explains the basis for greater EM activity in patients during DT, the authors suggest that when planning EM rehabilitation using an ET, the focus should first be on dynamic tasks and then, as EM control improves, the tasks should be expanded to include static elements.

## 6. Limitation

The study presented here has a limitation. Because of the high mortality rate and limited availability of participants meeting our inclusion criteria, one limitation is the small number of participants. Future studies could include larger numbers of patients to collect more data to confirm our findings. Moreover, future studies could obtain information about the neurological damage observed in the patients (occipital lobes, visual radiations). This will allow for a better understanding of the patients’ level of carrying out the researchers’ instructions and to what level the neurological damage could limit the ability to perform the tasks given to them.

## Figures and Tables

**Figure 1 ijerph-19-06280-f001:**
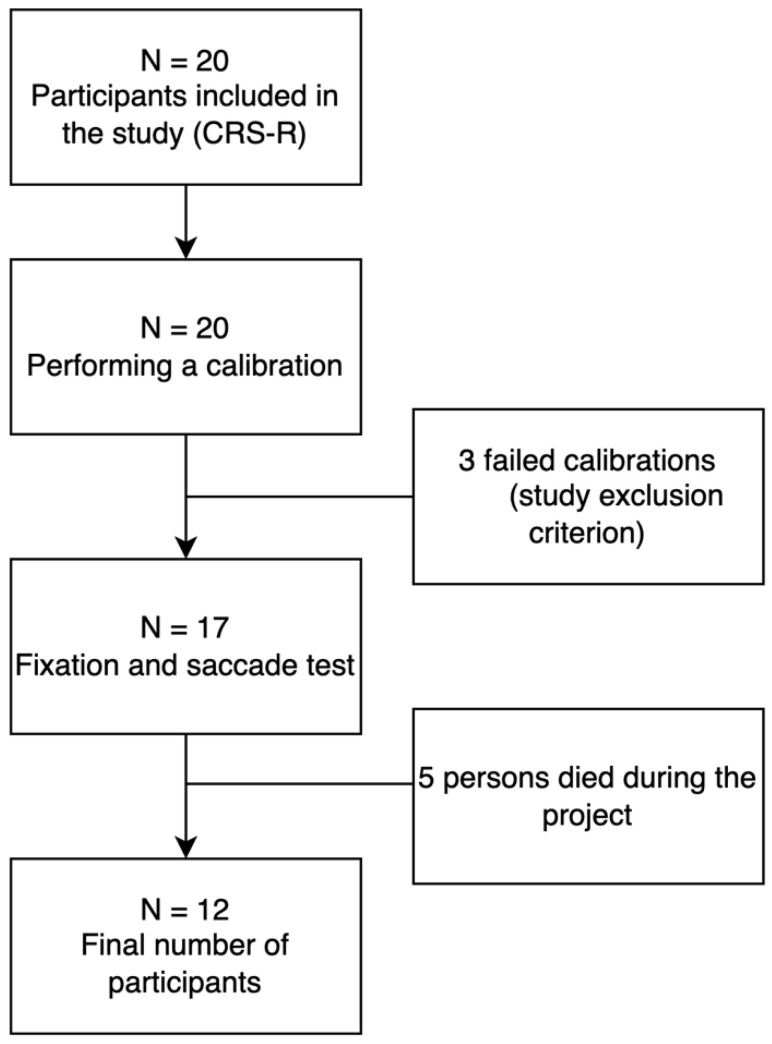
Flow chart. Patient recruitment process in the study.

**Figure 2 ijerph-19-06280-f002:**
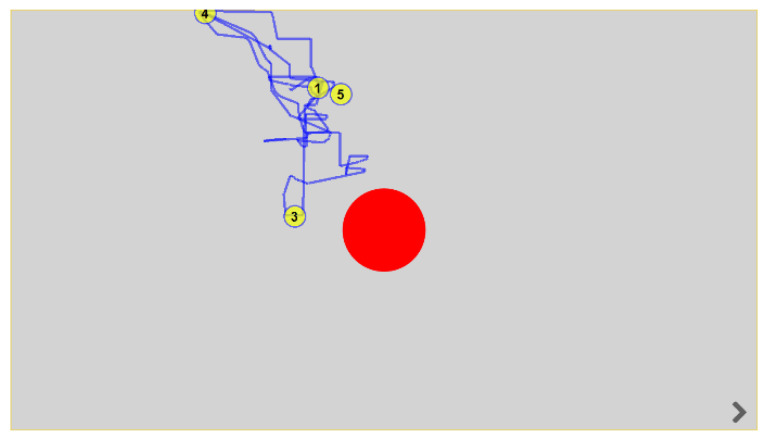
Image generated by the device after the test (ST) performed by P11. The red dot in the middle of the screen was the patient’s intended point of interest. Yellow dots mark the fixation points in the screen field.

**Figure 3 ijerph-19-06280-f003:**
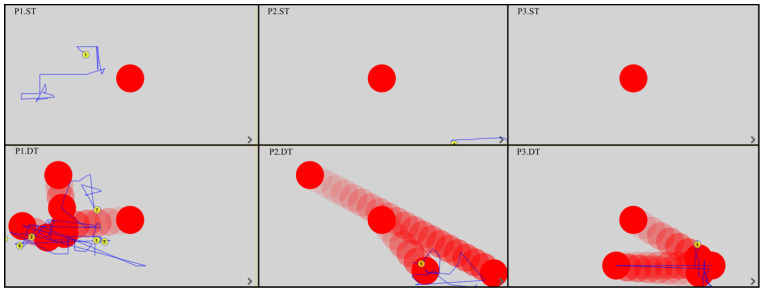
EM trajectories in P1–P3 in ST and DT (P—patient, ST—static task, DT—dynamic task).

**Figure 4 ijerph-19-06280-f004:**
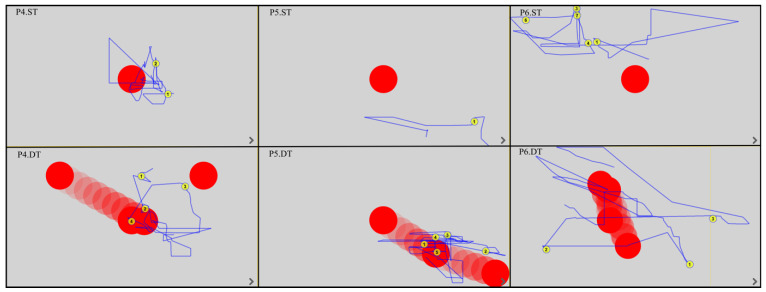
EM trajectories in P4–P6 in ST and DT (P—patient, ST—static task, DT—dynamic task).

**Figure 5 ijerph-19-06280-f005:**
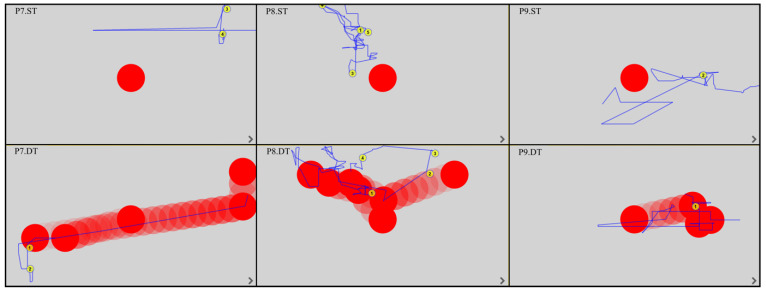
EM trajectories u P7–P9 in ST and DT (P—patient, ST—static task, DT—dynamic task).

**Figure 6 ijerph-19-06280-f006:**
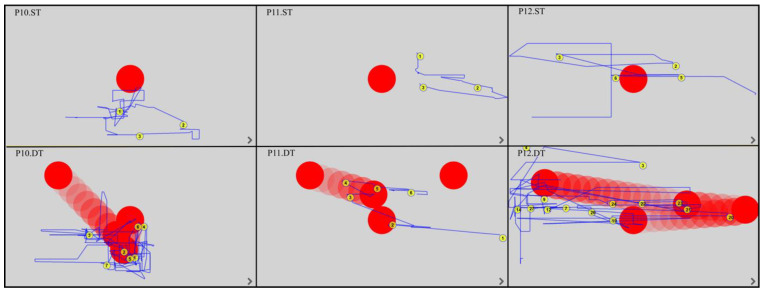
EM trajectories in P10–P12 in ST and DT (P—patient, ST—static task, DT—dynamic task).

**Table 1 ijerph-19-06280-t001:** Characteristics of the study group.

Sex		Female					Male						
Patient		P2	P3	P8	P10	P11	P1	P4	P5	P6	P7	P9	P12
Age		43	42	65	50	27	65	25	31	40	26	67	67
Diagnosis		ISBH	ILSS	HRSS	IRSS	CT	HRSS	CT	CT	CT	CT	BSS	HRSS
CRS-R	total	6	8	20	4	13	16	16	18	9	22	8	22
	subscales	(2/1/2/ 0/0/1)	(2/1/2/ 1/0/2)	(4/5/5/ 2/1/3)	(1/0/2/ 0/0/1)	(3/5/2/ 1/0/2)	(3/5/4/ 2/1/1)	(4/5/2/ 2/0/3)	(4/5/2/ 2/2/3)	(2/1/2/ 2/0/2)	(4/5/6/ 2/2/3)	(2/1/2/ 1/0/2)	(4/5/6/ 2/2/3)
CConscious state		UWS	UWS	eMCS	UWS	MCS	MCS	MCS	eMCS	UWS	eMCS	UWS	eMCS

Note: CT—Cerebrocranial Trauma, ILSS—Ischemic Left-Side Stroke; IRSS—Ischemic Right-Side Stroke; HRSS—Hemorrhagic Right-Sided Stroke; BSS—Brain Stem Stroke; ISBH—Ischemic Stroke in Both Hemispheres, P—Patient, MCS—Minimally Conscious State, UWS—Unresponsive Wakefulness Syndrome, eMCS—emergence from Minimally Conscious State; points in subscales are: auditory/visual/motor/oromotor-verbal/communication/arousal.

**Table 2 ijerph-19-06280-t002:** Number of on-screen and off-screen fixations by task type (ST and DT).

Patient	Fixation			
	In the Monitor Field	Outside the Monitor	In the Monitor Field	Outside the Monitor
	Type of Task			
	ST [N]		DT [N]	
P1	1	0	6	1
P2	1	3	1	0
P3	0	0	1	0
P4	2	0	4	0
P5	1	0	5	0
P6	5	2	3	0
P7	2	2	2	0
P8	4	1	4	0
P9	1	1	1	0
P10	3	0	7	0
P11	3	0	6	0
P12	4	2	14	12
**Me (Q1; Q3)**	2 (1; 3.5)	0.5 (0; 2)	4 (1.5; 6)	0 (0; 0)

Note: N—the number of fixations, ST—static task, DT—dynamic task, Me—median, Q—quartile.

**Table 3 ijerph-19-06280-t003:** Significance of differences between the number of fixations in static and dynamic tasks in and the out of the monitor area.

Pair of Variables	Wilcoxon Paired-Rank Order Test The Marked Results Are Significant with *p* < 0.5000			
	N	T	Z	*p*
ST-M and ST-OM	9	4.000000	2.191691	0.028403
DT-M and DT-OM	12	0.00	3.059412	0.002218
ST-M and DT-M	8	2.500000	2.170434	0.029975
ST-OM and DT-OM	7	9.000000	0.845154	0.398025

Note: ST-M—static task-fixations in monitor, DT-M—dynamic task-fixations in monitor, ST-OM—static task-fixations out of monitor, DT-OM—dynamic task-fixations out of monitor, T—the signed-rank sum, Z—Wilcoxon paired-rank order test, *p* <0.05000.

## Data Availability

The data presented in this study are available on request from the corresponding author.

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
