# Peer review of "Monitoring Eye Movements Depending on the Type of Visual Stimulus in Patients with Impaired Consciousness Due to Brain Damage"

_ijerph, 2022, doi:10.3390/ijerph19106280_

Round 1

Reviewer 1 Report

Thank you for the opportunity to review the second revision of this manuscript.  I appreciate the work you have put into making the requested clarifications.

Regrettably I do not read Polish, so I am uncertain about the response (2nd page) in Polish.

I suspect that many readers would share my questions regarding the degree to which these patients could follow commands, and the degree to which neurological lesions (occipital lobes, optic radiations) might limit the extent to which a patient could execute some of the tasks.  You mention in your responses that, "We will consider [such limitations] in further research."  In fairness to readers, it may be worthwhile stating this in the conclusions and limitations at the end of the paper.

Author Response

We would like to sincerely thank the Reviewer for the positive feedback and helpful comments. We strongly focused our efforts on the points made in your letter. We would like to respond to your opinion based on our careful revision, point by point below.

Reviewer: Regrettably I do not read Polish, so I am uncertain about the response (2nd page) in Polish.

Answer: I am very sorry for this oversight, once again I will send the translated part.

  • Answer (2stround): Thank you for your comment. The first number is for the auditory test. The following numbers refer to the subscale score: visual , motor, oromotor-verbal, communication and arousal. We decided to add to the description of the individual subscale items  of CRS-R under the table 1.

In CRS-R Administration and Scoring Guidelines (available online: https://www.sralab.org/rehabilitation-measures/coma-recovery-scale-revised)  describes in detail the procedure for taking the test, including the various subscales, the patient's expected response, and the number of points the patient earns based on his responses.

On the auditory subscale, the subject who received the 2 pkt is able to perform the task described in Guidelines ‘’ Standing behind the patient and out of view, present an auditory stimulus (e.g. patient’s name,voice, noise) from the right side for 5 seconds. Perform a second trial presenting the auditory stimulus from the left side. Repeat above procedure for a total of 4 trials, 2 on each side. If needed, reorient the head to midline between trials. The response expected from the patient is described as: „ Head and/or eyes orient toward the location of the stimulus on both

Assuming this fact, we can conclude that the patient understands what is being said to him and thus is able to follow other commands.

When it comes to the value of 2.9 In the case of the auditory subscale to which the reviewer refers, we would like to point out that it was not our intention to average the results of the individual subscales but only to present them openly in order to show information about the patients and the results they obtained in the individual subscales of the CRS-R. We therefore treat the score obtained on each subscale as a starting point for further analysis of patient performance with the eye tracker. Our experience with patients shows that there is often no direct transfer of behavioral scale scores to eye tracker results. This is also confirmed by Monti et all. 2010.

Reviewer: I suspect that many readers would share my questions regarding the degree to which these patients could follow commands, and the degree to which neurological lesions (occipital lobes, optic radiations) might limit the extent to which a patient could execute some of the tasks.  You mention in your responses that, "We will consider [such limitations] in further research."  In fairness to readers, it may be worthwhile stating this in the conclusions and limitations at the end of the paper.

Answer: Thank you for this suggestion; on the recommendation of the reviewer we have, of course, included such information in the manuscript, it can be found in the section limitations (see lines: 316-320) its wording is: Moreover, future studies could obtain information about the neurological damage observed in the patients (occipital lobes, visual radiations). This will allow for a better understanding of the patients' level of carrying out the researchers' instructions and to what level the neurological damage could limit the ability to perform the tasks given to them.

Reviewer 2 Report

1.Lack of quantitative comparative analysis with existing similar research work;

2. Unclear description of research contribution, innovation and motivation;

3. The evaluation index of the experimental results of the manuscript is not clear.

Author Response

We would like to sincerely thank the Reviewer for the positive feedback and helpful comments. We strongly focused our efforts on the points made in your letter. We would like to respond to your opinion based on our careful revision, point by point below.

Reviewer: 1. Lack of quantitative comparative analysis with existing similar research work;

Answer: thank you for that comment. Because of the number of participants, due mainly to the high mortality rate, we used statistics that were feasible with such a small group. It is difficult to make statistical comparisons of results that represent different aspects and are collected in different ways. In the initial part of the discussion (lines: 227-229) We point out that there is a lack of reports in the literature on the importance of the "type of stimulus" to provoke a response in a patient after severe brain damage and at the same time this would be work that uses objective research tools. For the example, we mention the works that we also refer to in the manuscript:

  1. Sun, Y.; Wang, J.; Heine, L.; Huang, W.; Wang, J.; Hu, N.; Hu, X.; Fang, X.; Huang, S.; Laureys, S.; et al. Personalized Objects Can Optimize the Diagnosis of EMCS in the Assessment of Functional Object Use in the CRS-R: A Double Blind, Randomized Clinical Trial. BMC Neurol 2018, 18, 38, doi:10.1186/s12883-018-1040-5.
  2. Stenberg, J.; Godbolt, A.K.; Möller, M.C. The Value of Incorporating Personally Relevant Stimuli into Consciousness Assessment with the Coma Recovery Scale - Revised: A Pilot Study. J Rehabil Med 2018, 50, 253–260, doi:10.2340/16501977-2309.

Even if in some articles authors examined pursuit in patients with disorders of consciousness using modern technology such as BCI. However, eg.  Xiao et al. 2018 focuses on patient evaluation without taking into account the type of stimulus that they are looking at.

  1. Xiao, Q. Xie, Q. Lin, T. Yu, R. Yu and Y. Li, "Assessment of Visual Pursuit in Patients With Disorders of Consciousness Based on a Brain-Computer Interface," in IEEE Transactions on Neural Systems and Rehabilitation Engineering, vol. 26, no. 6, pp. 1141-1151, June 2018, doi: 10.1109/TNSRE.2018.283581

Reviewer: 2. Unclear description of research contribution, innovation and motivation;

Answer: thank you for that comment. The main motivation of the study was to present the essence of the type of stimulus that could be used in the therapy of patients after brain damage. We have tried to emphasize this by adding information to the description in section introduction: it is necessary to consider the types of visual stimuli in eye movement therapy.

Our study is novel because there are no studies available in the literature databases showing how patients with disorders of consciousness (DOC) respond to the type of stimulus displayed with ET. Information about the novelty of the study is included in the section introduction (see lines:79-85).

Reviewer: 3. The evaluation index of the experimental results of the manuscript is not clear.

Answer: thank you for that comment. As we mentioned before we use statistical analysis possible to use in such small group. Could we please be more specific about what the reviewer meant by „evaluation index”. if there is a need to add to this information, we will try to do so.

Reviewer 3 Report

The author's research is a work of great medical and social value. The author compared the eye movement trajectories of different brain injury patients during saccade and gaze, and concluded that the subjects showed greater eye movement behavior when observing moving points. I think there are several points in the manuscript that need to be discussed:
1. In the introduction, it may not be necessary to spend too much time introducing the basic principles of eye movement.
2. As the author said in part 6, the scale of the subjects is too small to effectively support the research conclusions. If possible, the study needs to expand the number of subjects.
3. The most important point is that the final conclusion of this paper is "it is necessary to consider the types of visual stimuli in eye movement therapy". This conclusion is obvious. It is suggested that the author clarify the research conclusion.

Author Response

We would like to sincerely thank the Reviewer for the positive feedback and helpful comments. We strongly focused our efforts on the points made in your letter. We would like to respond to your opinion based on our careful revision, point by point below.

Reviewer: 1. In the introduction, it may not be necessary to spend too much time introducing the basic principles of eye movement.

Answer: thank you for that suggestion. We were very careful to present the problem of vision control as clearly as possible, and after revising the manuscript with the advice of the reviewer, we removed some of the text so that the flow of thought would be preserved.

Reviewer: 2. As the author said in part 6, the scale of the subjects is too small to effectively support the research conclusions. If possible, the study needs to expand the number of subjects.

Answer: thank you for that comment. Our experience shows that due to high mortality rate among patients with severe brain injury, it is very difficult to maintain a constant number of participants (this type of research is accompanied by a significant drop-out rate). Because of the limitation resulting from the size of the group, we decided to present the results obtained, recognizing their substantive value.

Reviewer: 3. The most important point is that the final conclusion of this paper is “it is necessary to consider the types of visual stimuli in eye movement therapy”. This conclusion is obvious. It is suggested that the author clarify the research conclusion.

Answer: thank you for that comment. Following the reviewer's advice, we have changed final part of the Abstract section as follows: The findings suggest that effective eye movement therapy must incorporate dynamic stimuli (lines: 18-19). Additionally, we have added one more sentence in the Conclusion section see lines (351-352).

Round 2

Reviewer 2 Report

My question was answered. The paper has improved.

This manuscript is a resubmission of an earlier submission. The following is a list of the peer review reports and author responses from that submission.

Round 1

Reviewer 1 Report

Thank you for the opportunity to review this manuscript.

The authors allude to several related, but distinct potential purposes of studying eye movements in patients with disorders of consciousness.  The first is diagnosis (“important in correctly diagnosing them,” lines 184-185).  The second is treatment (“eye movement therapy” in line 19; “eye movement rehabilitation” in lines 239-240).  The third is facilitating communication for a patient whose motor output is impaired (“increase the ability to communicate,” lines 231-233).  However, the manuscript seems undecided about which of these is being explored.  It would help the reader if the purpose was presented in a more focused fashion.

The manuscript’s conclusion (“the visual dynamic stimulus attracted the patient’s attention more”) seems justified, but this idea is not novel.  A general principle of sensory input is that biological sensors in any modality are better at detecting change (dynamic stimulation) than stasis (constant stimulation) (Gray, J.A.B.; Malcolm, J.L. The initiation of nerve impulses by mesenteric Pacinian corpuscles. Proc. R. Soc. London. Ser. B Biol. Sci. 1950, 137, 96–114).  The observation that this same principle applies in a diseased individual does not seem very remarkable, though the point is well-taken that when trying to elicit a response (for the purposes of diagnosis), it is helpful to insure that a sufficiently provocative stimulus is employed.  The authors need to demonstrate more clearly what is novel about their findings; simply stating that, “We are not aware if there were studies in the literature presenting analysis of eye movements in patients after brain injury with disorders of consciousness” (lines 84-85) is insufficient.

The technological description is inadequate; for example, few details are provided about the “eye tracker (ET)” device.

The presentation of the data is unclear in places.  For example, in table 1, it is unclear where the “Female” versus “Male” patient cutoff is.  It appears that the first five on the left (P2, P3, P8, P10, P11) are female, and the next seven (P1, P4, P5, P6, P7, P9, P12), but this might be made more clear if there was a dividing line between the two groups, or there was shading, or some other mechanism.

Some of the language is difficult to understand.  For instance, in lines 149-150 the authors state, “Each fixation located between 150 and 600 ms was automatically assigned a number by the program.”  Does this mean that if the duration of the fixation was 150 - 600 ms, then it was considered a fixation point worthy of being labeled?

I also have some methodological concerns about how the data were collected.

While this manuscript focuses on output from the patient, I do not see any clear process by which the investigators assessed the patient’s ability to comprehend instructions.  The authors state, “The patient was asked to observe…” (line 127).  How do we know that the patients understood the instructions adequately?  For that matter, how do we even know that the patients could hear the instructions?  Patients with significant head trauma often suffer ossicular chain dislocations, labyrinthine concussions, and traction injuries of the cochlear nerve.  Were any audiologic studies performed in these patients, at least those studies that do not involve voluntary responses (such as otoacoustic emissions or brainstem auditory evoked responses)?

I also do not see any clear process establishing the integrity of the patient’s visual input.  Do we know whether these patients have retinal damage or optic nerve damage?  For that matter, do we know which parts of the cerebral cortex are damaged?  The descriptions in the legend to Table 1 are rather vague (e.g., “Ischemic Right-Side Stroke” — does this mean all the cerebral lobes on the right side, or just some subset?).

Author Response

We would like to sincerely thank the Reviewer for the positive feedback and helpful comments. We strongly focused our efforts on the points made in your letter. We would like to respond to your opinion based on our careful revision, point by point below.

Reviewer: The authors allude to several related, but distinct potential purposes of studying eye movements in patients with disorders of consciousness.  The first is diagnosis (“important in correctly diagnosing them,” lines 184-185). 

 The second is treatment (“eye movement therapy” in line 19; “eye movement rehabilitation” in lines 239-240).  The third is facilitating communication for a patient whose motor output is impaired (“increase the ability to communicate,” lines 231-233).  However, the manuscript seems undecided about which of these is being explored.  It would help the reader if the purpose was presented in a more focused fashion.

Answer: Thank you for your comment. We agree that unclear statement of purpose makes the paper difficult to read and is needlessly confusing. So, after carefully reviewing your comment, we have re-read the text of the paper and sorted out purpose.

Modified paragraphs in the Discussion section are:

There are no studies to date showing how the type of stimulus displayed to patients with disorders of consciousness affects their interest in observing that stimulus. However, there have been studies on how patients respond to personalized and non-personalized visual stimuli. For this purpose, the CRS-R scale was used […]”

“The present study analyzed how patients with impaired consciousness resulting from brain injury observe the displayed point on the screen and which visual stimulus (static or dynamic) arouses more interest in what is happening on the monitor, as expressed by their EM control. Based on the presented results, it can be clearly concluded that the visual dynamic stimulus attracted the patient's attention more”

“We expect that the incorporation of dynamic stimuli into a rehabilitation program for people after brain damage will be beneficial not only because of the potential to improve EM control, but also to improve the interaction between the patient and the eye tracker”

Reviewer: The manuscript’s conclusion (“the visual dynamic stimulus attracted the patient’s attention more”) seems justified, but this idea is not novel.  A general principle of sensory input is that biological sensors in any modality are better at detecting change (dynamic stimulation) than stasis (constant stimulation) (Gray, J.A.B.; Malcolm, J.L. The initiation of nerve impulses by mesenteric Pacinian corpuscles. Proc. R. Soc. London. Ser. B Biol. Sci. 1950137, 96–114).  The observation that this same principle applies in a diseased individual does not seem very remarkable, though the point is well-taken that when trying to elicit a response (for the purposes of diagnosis), it is helpful to insure that a sufficiently provocative stimulus is employed.  The authors need to demonstrate more clearly what is novel about their findings; simply stating that, “We are not aware if there were studies in the literature presenting analysis of eye movements in patients after brain injury with disorders of consciousness” (lines 84-85) is insufficient.

Answer: Thank you for your comment. In some articles authors examined pursuit in patients with disorders of consciousness using modern technology such as BCI. However, eg.  Xiao et al. 2018 focuses on patient evaluation without taking into account the type of stimulus that they are looking at.

  1. Xiao, Q. Xie, Q. Lin, T. Yu, R. Yu and Y. Li, "Assessment of Visual Pursuit in Patients With Disorders of Consciousness Based on a Brain-Computer Interface," in IEEE Transactions on Neural Systems and Rehabilitation Engineering, vol. 26, no. 6, pp. 1141-1151, June 2018, doi: 10.1109/TNSRE.2018.283581

In a different example, results of other research that takes into account the type of stimulus is not performed with modern technology like eye tracking (eg. Sun, Y.; Wang, J.; Heine, L.; Huang, W.; Wang, J.; Hu, N.; Hu, X.; Fang, X.; Huang, S.; Laureys, S.; et al. Personalized Objects Can Optimize the Diagnosis of EMCS in the Assessment of Functional Object Use in the CRS-R: A Double Blind, Randomized Clinical Trial. BMC Neurol 2018, 18, 38, doi:10.1186/s12883-018-1040-5.)

Our study is novel because there are no studies available in the literature databases showing how patients with disorders of consciousness (DOC) respond to the type of stimulus displayed with ET. The modified part in the Introduction section is in lines 87-91: Studies on stimulus type selection have been conducted for a long time. Findings from animal studies in laboratory settings show that biological sensors are more sensitive to detecting dynamic than static stimuli (Gray, Malcolm 1950). However, there are no studies available in the literature databases showing how patients with disorders of consciousness (DOC) respond to the type of stimulus displayed with ET

Reviewer: The technological description is inadequate; for example, few details are provided about the “eye tracker (ET)” device.

Answer: Thank you for your comment. We added some technical data to the description; it can be found in the Research Tool section.

Reviewer: The presentation of the data is unclear in places.  For example, in table 1, it is unclear where the “Female” versus “Male” patient cutoff is.  It appears that the first five on the left (P2, P3, P8, P10, P11) are female, and the next seven (P1, P4, P5, P6, P7, P9, P12), but this might be made more clear if there was a dividing line between the two groups, or there was shading, or some other mechanism.

Answer: Thank you for your suggestion. We decided to divide these two groups by a vertical line.

Reviewer: Some of the language is difficult to understand.  For instance, in lines 149-150 the authors state, “Each fixation located between 150 and 600 ms was automatically assigned a number by the program.”  Does this mean that if the duration of the fixation was 150 - 600 ms, then it was considered a fixation point worthy of being labeled?

Answer: Thank you for your comment. The content we wanted to present is “[…] duration of the fixation was 150 - 600 ms, then it was considered a fixation point worthy of being labeled?” We decided to send the text to the linguistic specialist once again to correct some inaccuracies in the translation. 

Reviewer: While this manuscript focuses on output from the patient, I do not see any clear process by which the investigators assessed the patient’s ability to comprehend instructions.  The authors state, “The patient was asked to observe…” (line 127). How do we know that the patients understood the instructions adequately? 

Answer: Thank you very much for that question. We agree with the reviewer that it is difficult to definitively specify with severe brain damage whether the patient understands everything we are saying to them. However, based on our observations made while conducting the previous study in this group of patients, we note that the patient directs his gaze toward the projected image. We have presented this intentionality of choice in earlier publications in:

  1. Kujawa K, Zurek G, Kwiatkowska A, Olejniczak R, Zurek A: Assessment of language functions in patients with disorders of consciousness using and alternative communication tool. Frontiers in Neurology, 20 July 2021 https://doi.org/10.3389/fneur.2021.684362
  2. Kujawa K, Żurek A, Gorączko A, Zurek G. Application of high-tech solution for memory assessment in patients with disorders. Accepted to be published in Frontiers in Neurology. doi: 3389/fneur.2022.841095

Excerpt from the article 2: “[...] Additionally Friedman's ANOVA were performed on each patient's results to verify whether it is significantly higher than random level. We found significant (non-random) differences in the distributions between the categories of tasks that patients performed

Reviewer: For that matter, how do we even know that the patients could hear the instructions?  Patients with significant head trauma often suffer ossicular chain dislocations, labyrinthine concussions, and traction injuries of the cochlear nerve.  Were any audiologic studies performed in these patients, at least those studies that do not involve voluntary responses (such as otoacoustic emissions or brainstem auditory evoked responses)?

Answer: This reviewer question is a valuable cue for us to fill in some details and add data from the CRS-R subscales in the manuscript. Auditory testing is included in the CRS-R scale in the auditory functional scale subscale. None of our patients scored 0 in this category; therefore, patients were assumed to be hearing to facilitate the reader's understanding of our results, the point values obtained by patients on each subscale of the CRS-R have been added in Table 1.

Reviewer:  I also do not see any clear process establishing the integrity of the patient’s visual input.  Do we know whether these patients have retinal damage or optic nerve damage?  For that matter, do we know which parts of the cerebral cortex are damaged? 

Answer: One of the subscales of the CRS-R that determines the appropriate amount of points to assign to a patient, is the visual function scale. As mentioned above, detailed scores from each subscale are included in Table 1. In Poland, detailed ophthalmological examinations are not performed after brain injury. However, if the patient had partial or complete optic nerve damage, user could still control the device by recording movements from one eyeball. In the case of bilateral, complete optic nerve damage, it would not be possible to perform calibration on the device and thus include the patient in the project. The results of our previous studies in this group of patients indicate that they were able to solve the language function and memory tasks given to them at high and very high levels:

  1. Kujawa K, Zurek G, Kwiatkowska A, Olejniczak R, Zurek A: Assessment of language functions in patients with disorders of consciousness using and alternative communication tool. Frontiers in Neurology, 20 July 2021 https://doi.org/10.3389/fneur.2021.684362
  2. Kujawa K, Żurek A, Gorączko A, Zurek G. Application of high-tech solution for memory assessment in patients with disorders. Accepted to be published in Frontiers in Neurology. doi: 3389/fneur.2022.841095

Reviewer:  The descriptions in the legend to Table 1 are rather vague (e.g., “Ischemic Right-Side Stroke” — does this mean all the cerebral lobes on the right side, or just some subset?).

Answer: The term the reviewer is referring to (e.g. “Ischemic Right-Side Stroke”) is a typical type that is used in clinical practice. It gives preliminary information to doctor, therapist, or other health care provider about the expected neurological consequences of this injury. For the purposes of this study, however, we did not further distinguish in which lobe the lesion focus was located.

Reviewer 2 Report

I consider that this manuscript should be rewritten again as a pilot study. How have you calculated the sample size?
With that low number of patients you cannot draw generalized conclusions.
Furthermore, in the discussion, the authors should compare their results with other studies.

Author Response

We would like to sincerely thank the Reviewer for the feedback and helpful comments. We strongly focused our efforts on the points made in your letter. We would like to respond to your opinion based on our careful revision, point by point below.

Reviewer: I consider that this manuscript should be rewritten again as a pilot study. How have you calculated the sample size?

Answer: Thank you for your question. As presented in the Materials and Methods section, 20 subjects with different etiologies of brain injury were included in the study. For this reason, our study was designed as qualitative.

Reviewer: With that low number of patients you cannot draw generalized conclusions.

Answer: Thank you for your comment.  We are aware that a larger number of subjects can minimize false-positive results.  We considered this as a limitation in our study (see lines 422-426)

Reviewer: Furthermore, in the discussion, the authors should compare their results with other studies.

Answer: There are no studies available in the literature databases showing how patients with Disorders of Consciousness (DOC) respond to the type of stimulus (static vs dynamic) displayed using Eye tracking. For this reason, it was difficult for us to compare our own findings with those reported in other publications.  Where it seemed possible, we have tried to present the results obtained in our study in the light of other available studies.

Reviewer 3 Report

This is a great study. I would accept this paper after these revisions:

  1. How did you extract the eye movement data? Somehow I want to see the flow. See this study:
  • Lin, C. J.; Widyaningrum, R. Eye pointing in stereoscopic displays. Journal of Eye Movement Research 2016, 9.
  1. How did you make sure that those 12 participants had equal level of brain damage? I believe it is the limitation of the study and need to be mentioned before the conclusion.
  2. Lin et al (2019) and Lin et al (2018) proved that several eye movement measures had correlations with symptoms. You might cite these studies:
  • Lin, C. J.; Prasetyo, Y. T.; Widyaningrum, R. Eye movement measures for predicting eye gaze accuracy and symptoms in 2D and 3D displays. Displays 2019, 60, 1–8.
  • Lin, C. J.; Prasetyo, Y. T.; Widyaningrum, R. Eye movement parameters for performance evaluation in projection-based stereoscopic display. Journal of Eye Movement Research 2018, 11.
  1. There are many eye movement measures. This study only collected fixation inside and outside the monitor. In fact, there are many eye movement measures such as number of fixation, fixation duration, time to first fixation, and even pupil size. Again, this study need to mention it in the limitations.
  2. I believe this study need to add statistical analysis to prove the significance of the result. At least ANOVA.
  3. Did you collect any subjective discomfort data during the experiment?

Author Response

We would like to sincerely thank the Reviewer for the positive feedback and helpful comments. We strongly focused our efforts on the points made in your letter. We would like to respond to your opinion based on our careful revision, point by point below.

Reviewer: How did you extract the eye movement data?

Answer: Images were automatically generated by the device and downloaded in the format .png

Reviewer: How did you make sure that those 12 participants had equal level of brain damage? I believe it is the limitation of the study and need to be mentioned before the conclusion.

Answer: Thank you for your question. We see the number of participants as a limitation, however, even if all patients were post-traumatic brain injury (CT) it's highly possible that each would still have a different clinical manifestation. Thus, it can be said that two identical damages are very rare. Because of the broad spectrum of clinical manifestations in neurological patients, we decided to not consider the etiology of brain damage and to include patients according to the designated inclusion criteria.

Reviewer: Lin et al (2019) and Lin et al (2018) proved that several eye movement measures had correlations with symptoms. You might cite these studies:

  1. Lin, C. J.; Prasetyo, Y. T.; Widyaningrum, R. Eye movement parameters for performance evaluation in projection-based stereoscopic display. Journal of Eye Movement Research 2018, 11.
  2. Lin, C. J.; Prasetyo, Y. T.; Widyaningrum, R. Eye movement measures for predicting eye gaze accuracy and symptoms in 2D and 3D displays. Displays 2019, 60, 1–8.

Answer: Thank you for suggesting the above literature. We went through the both very interesting publications but it was very difficult for us to relate to it due to the fact that the study group consisted of healthy subjects without visual impairment. Quoting from the article above „All participants had normal or corrected to normal visual acuity (1.0 in the decimal unit)”

There is a part in article (Lin, 2019) in which the author cites someone else's research that does not clearly define the relationship of stroke to selected measures of eye movement. „In addition, this model could not explain further about the effect of stroke complicacy on selected eye movement measures since the interrelationship among factors were not analyzed further. Yu et al.”

The term symptoms in this article does not refer to symptoms associated with a disease but to commonly used terms in VR „side effects” (Lin, 2019) this term is not clearly developed. At work D.N. Hoffman, A.R. Girshick, K. Akeley, M.S. Banks, Vergence-accommodation conflicts hinder visual performance and cause visual fatigue, J. Vis. 8 (3) (2008) 1–30. The term symptoms refers to discomfort and tiredness.

Reviewer: There are many eye movement measures. This study only collected fixation inside and outside the monitor. In fact, there are many eye movement measures such as number of fixation, fixation duration, time to first fixation, and even pupil size. Again, this study need to mention it in the limitations.

Answer: Thank you for this comment. There are indeed many different measures of eye movement that the reviewer mentioned. The device we used in this study operates by emitting infrared radiation that creates reflections on the corneas of the eyes. This allows them to become reference points for the center of the pupil. The deformations of the reflections called glints are then analyzed. Then a special algorithm makes it possible to relate the eye position to the fixation point on the monitor. Information about the so-called First Purkinje Point and its relation to glints has been included in the Research Tool section. We also did not take pupil size into account due to the fact that the operating system of the eye tracker used in this study does not respond to changes in size, and thus this is not "pupil tracking." We also omitted measuring time to first fixation because it was not of interest.

Reviewer: I believe this study need to add statistical analysis to prove the significance of the result. At least ANOVA.

Answer: For this type of data, parametric analyses based on mean and standard deviation could not be performed because the variables are not continuous, they are measured on short scales and one of the objects clearly stands out from the others. Thus, analyses based on rank tests seem most appropriate. Therefore, statistical description was based on positional measures, i.e. minimum, maximum, quartiles and median, and then comparing the compatibility of the distributions of the studied variables using Friedman rank test (ANOVA).

Reviewer: Did you collect any subjective discomfort data during the experiment?

Answer: In this study, we found no discomfort in the participating patients due to the fact that the duration of the test they performed was relatively short (10 seconds to perform calibration and 20 seconds total to perform 2 tasks). However, in our other study with the eye tracker, we observed that prolonged time with the device can cause a tiredness that results in the need for rest.

Round 2

Reviewer 1 Report

We would like to sincerely thank the Reviewer for the positive feedback and helpful comments. We strongly focused our efforts on the points made in your letter. We would like to respond to your opinion based on our careful revision, point by point below.

Reviewer (1st round): The authors allude to several related, but distinct potential purposes of studying eye movements in patients with disorders of consciousness.  The first is diagnosis (“important in correctly diagnosing them,” lines 184-185).  The second is treatment (“eye movement therapy” in line 19; “eye movement rehabilitation” in lines 239-240).  The third is facilitating communication for a patient whose motor output is impaired (“increase the ability to communicate,” lines 231-233).  However, the manuscript seems undecided about which of these is being explored.  It would help the reader if the purpose was presented in a more focused fashion.

Answer: Thank you for your comment. We agree that unclear statement of purpose makes the paper difficult to read and is needlessly confusing. So, after carefully reviewing your comment, we have re-read the text of the paper and sorted out purpose.

Modified paragraphs in the Discussion section are:

There are no studies to date showing how the type of stimulus displayed to patients with disorders of consciousness affects their interest in observing that stimulus. However, there have been studies on how patients respond to personalized and non-personalized visual stimuli. For this purpose, the CRS-R scale was used […]”

“The present study analyzed how patients with impaired consciousness resulting from brain injury observe the displayed point on the screen and which visual stimulus (static or dynamic) arouses more interest in what is happening on the monitor, as expressed by their EM control. Based on the presented results, it can be clearly concluded that the visual dynamic stimulus attracted the patient's attention more”

“We expect that the incorporation of dynamic stimuli into a rehabilitation program for people after brain damage will be beneficial not only because of the potential to improve EM control, but also to improve the interaction between the patient and the eye tracker”

Reviewer (2nd round): Thank you for the clarification.

Reviewer (1st round): The manuscript’s conclusion (“the visual dynamic stimulus attracted the patient’s attention more”) seems justified, but this idea is not novel.  A general principle of sensory input is that biological sensors in any modality are better at detecting change (dynamic stimulation) than stasis (constant stimulation) (Gray, J.A.B.; Malcolm, J.L. The initiation of nerve impulses by mesenteric Pacinian corpuscles. Proc. R. Soc. London. Ser. B Biol. Sci. 1950137, 96–114).  The observation that this same principle applies in a diseased individual does not seem very remarkable, though the point is well-taken that when trying to elicit a response (for the purposes of diagnosis), it is helpful to insure that a sufficiently provocative stimulus is employed.  The authors need to demonstrate more clearly what is novel about their findings; simply stating that, “We are not aware if there were studies in the literature presenting analysis of eye movements in patients after brain injury with disorders of consciousness” (lines 84-85) is insufficient.

Answer: Thank you for your comment. In some articles authors examined pursuit in patients with disorders of consciousness using modern technology such as BCI. However, eg.  Xiao et al. 2018 focuses on patient evaluation without taking into account the type of stimulus that they are looking at.

  1. Xiao, Q. Xie, Q. Lin, T. Yu, R. Yu and Y. Li, "Assessment of Visual Pursuit in Patients With Disorders of Consciousness Based on a Brain-Computer Interface," in IEEE Transactions on Neural Systems and Rehabilitation Engineering, vol. 26, no. 6, pp. 1141-1151, June 2018, doi: 10.1109/TNSRE.2018.283581

In a different example, results of other research that takes into account the type of stimulus is not performed with modern technology like eye tracking (eg. Sun, Y.; Wang, J.; Heine, L.; Huang, W.; Wang, J.; Hu, N.; Hu, X.; Fang, X.; Huang, S.; Laureys, S.; et al. Personalized Objects Can Optimize the Diagnosis of EMCS in the Assessment of Functional Object Use in the CRS-R: A Double Blind, Randomized Clinical Trial. BMC Neurol 2018, 18, 38, doi:10.1186/s12883-018-1040-5.)

Our study is novel because there are no studies available in the literature databases showing how patients with disorders of consciousness (DOC) respond to the type of stimulus displayed with ET. The modified part in the Introduction section is in lines 87-91: Studies on stimulus type selection have been conducted for a long time. Findings from animal studies in laboratory settings show that biological sensors are more sensitive to detecting dynamic than static stimuli (Gray, Malcolm 1950). However, there are no studies available in the literature databases showing how patients with disorders of consciousness (DOC) respond to the type of stimulus displayed with ET

Reviewer (2nd round): Is it true that no studies in the literature describe eye tracking in disorders of consciousness?  See:

Vanhaudenhuyse A, Schnakers C, Brédart S, Laureys S.  Assessment of visual pursuit in post-comatose states: use a mirror.  J Neurol Neurosurg Psychiatry. 2008 Feb;79(2):223. doi: 10.1136/jnnp.2007.121624.  PMID: 18202215 No abstract available.

Candelieri A, Cortese MD, Dolce G, Riganello F, Sannita WG.  Visual pursuit: within-day variability in the severe disorder of consciousness.  J Neurotrauma. 2011 Oct;28(10):2013-7. doi: 10.1089/neu.2011.1885.  PMID: 21770758

Dolce G, Lucca LF, Candelieri A, Rogano S, Pignolo L, Sannita WG.  Visual pursuit in the severe disorder of consciousness.  J Neurotrauma. 2011 Jul;28(7):1149-54. doi: 10.1089/neu.2010.1405. Epub 2011 Apr 21.  PMID: 21175278

Wannez S, Vanhaudenhuyse A, Laureys S, Brédart S.  Mirror efficiency in the assessment of visual pursuit in patients in minimally conscious state.  Brain Inj. 2017;31(11):1429-1435. doi: 10.1080/02699052.2017.1376755. Epub 2017 Oct 5.  PMID: 28980847

Reviewer (1st round): The technological description is inadequate; for example, few details are provided about the “eye tracker (ET)” device.

Answer: Thank you for your comment. We added some technical data to the description; it can be found in the Research Tool section.

Reviewer (2nd round): If an investigator wants to try to replicate this study, or apply the method to a new study, then they will need more detail on how to do this.  For example, later in the study you refer to the Statistica software package; that is the level of detail that a colleague would like to see.  What hardware and software were you using?

Reviewer (1st round): The presentation of the data is unclear in places.  For example, in table 1, it is unclear where the “Female” versus “Male” patient cutoff is.  It appears that the first five on the left (P2, P3, P8, P10, P11) are female, and the next seven (P1, P4, P5, P6, P7, P9, P12), but this might be made more clear if there was a dividing line between the two groups, or there was shading, or some other mechanism.

Answer: Thank you for your suggestion. We decided to divide these two groups by a vertical line.

Reviewer (1st round): Some of the language is difficult to understand.  For instance, in lines 149-150 the authors state, “Each fixation located between 150 and 600 ms was automatically assigned a number by the program.”  Does this mean that if the duration of the fixation was 150 - 600 ms, then it was considered a fixation point worthy of being labeled?

Answer: Thank you for your comment. The content we wanted to present is “[…] duration of the fixation was 150 - 600 ms, then it was considered a fixation point worthy of being labeled?” We decided to send the text to the linguistic specialist once again to correct some inaccuracies in the translation. 

Reviewer (1st round): While this manuscript focuses on output from the patient, I do not see any clear process by which the investigators assessed the patient’s ability to comprehend instructions.  The authors state, “The patient was asked to observe…” (line 127). How do we know that the patients understood the instructions adequately? 

Answer: Thank you very much for that question. We agree with the reviewer that it is difficult to definitively specify with severe brain damage whether the patient understands everything we are saying to them. However, based on our observations made while conducting the previous study in this group of patients, we note that the patient directs his gaze toward the projected image. We have presented this intentionality of choice in earlier publications in:

  1. Kujawa K, Zurek G, Kwiatkowska A, Olejniczak R, Zurek A: Assessment of language functions in patients with disorders of consciousness using and alternative communication tool. Frontiers in Neurology, 20 July 2021 https://doi.org/10.3389/fneur.2021.684362
  2. Kujawa K, Żurek A, Gorączko A, Zurek G. Application of high-tech solution for memory assessment in patients with disorders. Accepted to be published in Frontiers in Neurology. doi: 3389/fneur.2022.841095

Excerpt from the article 2: “[...] Additionally Friedman's ANOVA were performed on each patient's results to verify whether it is significantly higher than random level. We found significant (non-random) differences in the distributions between the categories of tasks that patients performed

Reviewer: For that matter, how do we even know that the patients could hear the instructions?  Patients with significant head trauma often suffer ossicular chain dislocations, labyrinthine concussions, and traction injuries of the cochlear nerve.  Were any audiologic studies performed in these patients, at least those studies that do not involve voluntary responses (such as otoacoustic emissions or brainstem auditory evoked responses)?

Answer: This reviewer question is a valuable cue for us to fill in some details and add data from the CRS-R subscales in the manuscript. Auditory testing is included in the CRS-R scale in the auditory functional scale subscale. None of our patients scored 0 in this category; therefore, patients were assumed to be hearing to facilitate the reader's understanding of our results, the point values obtained by patients on each subscale of the CRS-R have been added in Table 1.

Reviewer (2nd round): In the modified table I assume that the first digit pertains to the auditory function scale of the CSR‑R.  If that assumption is correct, then the average score on the auditory scale for these 12 patients is 2.9, which should mean that on average these patients can “localize to sound.”  Is that sufficient to conclude that they can follow instructions? 

Reviewer (1st round):  I also do not see any clear process establishing the integrity of the patient’s visual input.  Do we know whether these patients have retinal damage or optic nerve damage?  For that matter, do we know which parts of the cerebral cortex are damaged? 

Answer: One of the subscales of the CRS-R that determines the appropriate amount of points to assign to a patient, is the visual function scale. As mentioned above, detailed scores from each subscale are included in Table 1. In Poland, detailed ophthalmological examinations are not performed after brain injury. However, if the patient had partial or complete optic nerve damage, user could still control the device by recording movements from one eyeball. In the case of bilateral, complete optic nerve damage, it would not be possible to perform calibration on the device and thus include the patient in the project. The results of our previous studies in this group of patients indicate that they were able to solve the language function and memory tasks given to them at high and very high levels:

  1. Kujawa K, Zurek G, Kwiatkowska A, Olejniczak R, Zurek A: Assessment of language functions in patients with disorders of consciousness using and alternative communication tool. Frontiers in Neurology, 20 July 2021 https://doi.org/10.3389/fneur.2021.684362
  2. Kujawa K, Żurek A, Gorączko A, Zurek G. Application of high-tech solution for memory assessment in patients with disorders. Accepted to be published in Frontiers in Neurology. doi: 3389/fneur.2022.841095

Reviewer (2nd round): In the modified table I assume that the second digit pertains to the visual function scale of the CSR‑R.  If that assumption is correct, then one patient scored zero (no visual function) and four patients scored 1 (visual startle).  Is that adequate to conclude that their visual pathways are sufficiently intact to perform this task?

Reviewer (1st round):  The descriptions in the legend to Table 1 are rather vague (e.g., “Ischemic Right-Side Stroke” — does this mean all the cerebral lobes on the right side, or just some subset?).

Answer: The term the reviewer is referring to (e.g. “Ischemic Right-Side Stroke”) is a typical type that is used in clinical practice. It gives preliminary information to doctor, therapist, or other health care provider about the expected neurological consequences of this injury. For the purposes of this study, however, we did not further distinguish in which lobe the lesion focus was located.

Reviewer (2nd round): This uncertainty would make it difficult to interpret the results.  For example, if a stroke patient had involvement of the occipital lobe or optic radiations, then a test would not be able to conclude whether failure of pursuit was due to a disturbance of pursuit mechanisms (frontal and parietal lobes), or failure to see (occipital lobe and optic radiations).

Reviewer 2 Report

I consider that the methodology of this manuscript should be improved. Unfortunately, in the current state and since the authors have not followed my recommendation, it is not considered suitable for publication

Reviewer 3 Report

Great work. I would recommend this study be published as soon as possible. Congratulations!